# Virological outcomes of antiretroviral therapy and its determinants among HIV patients in Ethiopia: Implications for achieving the 95–95–95 target

**Tegene Atamenta Kitaw**[ID]*, **Ribka Nigatu Haile**

Department of Nursing, College of Health Science, Woldia University, Woldia, Ethiopia

* tegene2013@gmail.com

**Data Availability Statement:** The data for this study are available in the PHIA Project: (https://phia.icap.columbia.edu/).

## Abstract

### Background

Despite significant advancements in HIV treatment, virological outcomes remain a critical issue. Ethiopia did not meet the 90:90:90 targets set for 2020, which aimed for 90% of people on antiretroviral therapy to achieve viral suppression. As the country shifts its focus toward the 95:95:95 targets for 2030—seeking to achieve 95% viral suppression among those on ART—it is crucial to deepen our understanding of viral suppression and the factors that influence it.

### Methods

Virological suppression was examined among 410 HIV-positive individuals on ART using the EPHIA survey. The survey employed a two-stage, stratified sampling design across urban areas in nine regions and two city administrations. Data analysis was conducted with STATA version 18, and multicollinearity was assessed using variance inflation factors. A logistic regression model identified significant determinants of viral suppression, with variables having a p-value of $\leq 0.05$ considered statistically significant in the multivariable analysis.

### Results

The findings revealed that 364 participants (88.78%) achieved viral suppression. Key determinants of virological outcomes included a higher wealth level (AOR = 2.67, 95% CI: 1.15–6.22), the presence of active TB (AOR = 0.27, 95% CI: 0.14–0.57), hepatitis B virus (AOR = 0.20, 95% CI: 0.10–0.31), and the utilization of HIV support group care (AOR = 3.14, 95% CI: 1.35–6.30).

### Conclusion

Viral suppression among HIV patients is 88.78%, which even falls short of the WHO's 90% target for 2020, indicating the substantial work required to achieve 95% by 2030. To improve

**Funding:** The author(s) received no specific funding for this work.

**Competing interests:** The authors have declared that no competing interests exist.

virological outcomes, it is crucial to increase support for low-income patients, enhance management of co-infections like TB and hepatitis B, and expand access to HIV support groups for better adherence and care.

## Introduction

Human Immunodeficiency Virus (HIV) continue to be significant global health challenges, with Sub-Saharan Africa bearing the brunt of the epidemic [1]. In Ethiopia, it is also a public health concern, has made substantial progress in expanding access to antiretroviral therapy (ART). Despite these advances, virological outcomes remain a critical concern [2]. Although ART is effective in suppressing HIV replication and preventing disease progression, a substantial proportion of patients in Ethiopia experience incomplete viral suppression, reflecting a complex interplay of factors influencing treatment success [3].

While ART is a cornerstone of HIV management, its efficacy can be compromised by a range of determinants. Adherence to ART, a critical factor for achieving viral suppression, is often undermined by issues such as medication side effects, complex treatment regimens, and socio-economic barriers [4]. Additionally, the emergence of drug-resistant HIV strains further complicates treatment, necessitating more complex and expensive regimens [5]. Socio-economic factors, including poverty, limited access to healthcare facilities, and stigma associated with HIV, also play significant roles in affecting treatment outcomes [6]. Inadequate virological suppression has severe consequences for individual health and public health efforts. Persisting viral replication increases the risk of progression to AIDS, heightens susceptibility to opportunistic infections, and contributes to elevated mortality rates [7]. Furthermore, inadequate viral suppression poses challenges to the broader goal of HIV prevention, as it contributes to ongoing transmission and undermines efforts to control the epidemic [8]. The financial burden on the healthcare system is also significant, as managing complications and advanced disease requires additional resources and infrastructure [9].

Despite existing evidence, over four million individuals on antiretroviral therapy (ART) had detectable viral loads, and approximately one million HIV-related deaths occurred worldwide in 2017 [10]. Most patients on ART with detectable viral loads were found to harbor drug-resistant mutations. Studies have indicated that over 60% of these patients had resistance to at least one antiretroviral (ARV) drug [11–13]. Based on previous evidence, overall ART treatment failure rates were 15.9%, with virological failure rates at 6.3% [14].

The magnitude of HIV in Ethiopia presents a significant public health challenge, despite extensive ART programs. Addressing the determinants of virological outcomes is crucial for optimizing ART and combating the epidemic. This study aims to investigate the virological outcomes of ART among HIV patients in Ethiopia and identify key factors affecting these outcomes. By understanding these factors, the research seeks to improve ART efficacy, enhance patient adherence, and inform public health policies, ultimately aiming to reduce transmission rates and improve health outcomes for patients and communities in Ethiopia.

## Methods

### Study setting, study period and data source

According to forecasts from Trading Economics and recent census data, Ethiopia's total population reached 115.0 million by 2020 [15]. The data for this study is sourced from the Ethiopia Population-Based HIV Impact Assessment (EPHIA). Conducted between October 2017 and

April 2018, EPHIA is a national household survey designed to assess Ethiopia's response to its urban HIV epidemic. The survey provided household-based HIV counseling and testing, returned results, and referred those who tested positive to clinics. It also collected information on the uptake of HIV prevention, care, and treatment services, and estimated HIV-related parameters at the population level for individuals aged 0–64 years.

## Data extraction, population and eligibility

First, the project proposal was submitted to the EPHIA Surveys Program. After a thorough review, the EPHIA program approved the proposal and provided access to the survey datasets via an approval email. Data extraction was performed to identify HIV patients on ART aged 15–64 years, and this process took place from April 1 to April 30, 2024. The source population consisted of all HIV patients aged 15–64 years on ART, while the study population included those within this age range on ART in the selected enumeration area. Only patients who had been on ART for at least six months were included in the study.

## Sample size and sampling procedures

The EPHIA survey utilized a two-stage, stratified sampling design, covering urban areas across nine regions and two city administrations. In the first stage, 393 enumeration areas (EAs) were included. In the second stage, 30 households were randomly selected from each EA, resulting in a total of 11,810 households. Detailed sampling procedures are outlined in the final EPHIA survey report. Following household selection, the head of each household provided written consent for their household members to participate in the survey. Of the 20,170 adults who took part in the survey, 19,136 consented to HIV testing. For this study, we focused on HIV-positive participants aged 15–64. Among the 19,136 individuals who consented to HIV testing, 614 tested positives for HIV, with 410 of these individuals having been on ART for at least six months. Participants aged 15–64 years gave written consent via a tablet for an interview and for participation in the biomarker component of the survey, which included home-based testing, counseling, and the return of HIV test results, as well as consent for future research. Consequently, the study included 410 HIV-positive individuals who had been on ART for a minimum of six months.

## Study variables

The dependent variable is viral load suppression. This study considered different independent variables that are found in EPHIA survey datasets *(Table 1)*.

## Definitions

Viral load suppression is measured as the viral load of less than 1000 copies/mL of blood after minimum of six month of taking antiretroviral therapy [18].

## Data processing and analysis

The data were coded, cleaned, and edited. The data were analyzed using STATA. Categorical data were computed in terms of frequency distribution. Continuous data were presented through basic descriptive analyses by computing central tendency and dispersion. The outcomes of the participants were dichotomized into suppressed and not suppressed. A logistic regression model was computed to identify the significant determinants of viral load suppression. Explanatory variables with a p-value of $\leq 0.25$ in the bivariate analysis were included in the multivariable logistic regression model. Multicollinearity was checked by computing

**Table 1. List of independent variables for the assessment of virological outcomes of antiretroviral therapy and its determinants among HIV patients in Ethiopia.**

| Variable | Descriptions (classification |
|---|---|
| Age in year | <25, 25–45 and >45 |
| Sex | Male and female |
| Marital status | Married and not married |
| Education level | No education, primary and secondary and above |
| Region | Larger central: Tigray, Amhara, Oromia, SNNPR |
| | Small peripherals: Benishangul, Gambela, Afar, Somali |
| | Metropolis: Harari, Addis Ababa, Dire Dawa [16, 17]. |
| Wealth level | Poor, middle and rich |
| Alcohol consumption | Yes and no |
| Time to reach to HIV care service | Less than 1 hr and 1hr. and above |
| ART duration | Less than 12 months, 12–24 month and above 24 months |
| Active TB | No, yes |
| Active Hepatitis B virus | Negative and positive |
| HIV support group care utilization | No and yes |

variance inflation factors (VIF) and tolerance values. In the multivariable analysis, variables with a p-value of $\leq 0.05$ were considered statistically significant. COR and AOR with 95% confidence intervals and p-values were used to assess the strength of the relationships and determine statistically significant factors associated with the dependent variable. Finally, the findings were presented in text, tables, and graphs.

## Ethical considerations

The survey protocol was approved by the Institutional Review Boards of the Ethiopian Public Health Institute (EPHI, Ethiopia), Centers for Disease Control and Prevention (Atlanta, GA USA), and Columbia University (New York, NY USA). The EPHIA Data Analysis Advisory Committee at the EPHI approved the analysis of the data.

## Results

### Descriptive statistics of study participants

The median age of the participants was 37 years ±10.13 (SD). The majority of the participants (76.83%) were female. Regarding educational level, 19.76% of the participants had not attended formal education. Concerning comorbidity status, 26.10% of the participants had active TB, and 5.37% had hepatitis B. A total of 36.10% of the patients utilized HIV support group care services *(Table 2).*

### Virological outcomes of antiretroviral therapy

Out of the 410 HIV patients, 364 (88.78%) successfully achieved viral suppression, while 46 (11.22%) did not achieve suppression *(Fig 1).*

### Multicollinearity test

The variance inflation factor (VIF) and tolerance values were used to check for the existence of multicollinearity between variables. A VIF above 4 or a tolerance below 0.25 indicates potential multicollinearity. In this study, the maximum VIF was 1.17, the mean VIF was 1.07, and the minimum tolerance value was 0.86. Thus, there is no multicollinearity between covariates *(Table 3).*

**Table 2. Descriptive statistics of HIV patients on antiretroviral therapy in Ethiopia.**

| Variables | Categories | Viral suppression status | | Total (%) |
|---|---|---|---|---|
| | | Not suppressed (%) | Suppressed (%) | |
| **Age in year** | <25 | 7 (1.71%) | 22 (5.37%) | 29 (7.07%) |
| | 25–45 | 35 (8.54%) | 262 (63.90%) | 297 (72.44%) |
| | >45 | 4 (0.985) | 88 (19.51%) | 84 (20.49%) |
| **Sex** | Male | 6 (1.46%) | 89 (21.71%) | 95 (23.17%) |
| | Female | 40 (9.765) | 275 (67.07%) | 315 (76.83%) |
| **Marital status** | Married | 16 (3.90%) | 179 (43.66%) | 195 (47.56%) |
| | Not married | 30 (7.32%) | 185 (45.12%) | 215 (52.44%) |
| **Education level** | No education | 11 (2.68%) | 70 (17.07%) | 81 (19.76%) |
| | Primary | 23 (5.61%) | 204 (49.76%) | 227 (55.37%) |
| | Secondary and above | 12 (2.935) | 90 (21.95%) | 102 (24.88%) |
| **Region** | Larger central | 27(6.59%) | 217(52.93%) | 244(59.51%) |
| | Small peripherals | 5(1.22%) | 64(15.61%) | 69(16.83%) |
| | Metropolis | 14(3.41%) | 83(20.24%) | 97(23.66%) |
| **Wealth level** | Poor | 23 (5.61%) | 106 (25.85%) | 129 (31.46%) |
| | Middle | 9 (2.20%) | 104 (25.37%) | 113 (27.56%) |
| | Rich | 14 (3.41%) | 154 (37.56%) | 168 (40.98%) |
| **Alcohol consumption** | Yes | 14 (3.415) | 112 (27.32%) | 126 (30.73%) |
| | No | 32 (7.80%) | 252 (61.46%) | 284 (69.27%) |
| **Time to reach to HIV care service** | Less than 1 hr. | 40 (9.76%) | 303 (73.90%) | 343 (83.66%) |
| | 1hr. and above | 6 (1.46%) | 61 (14.88%) | 67 (16.34%) |
| **ART duration** | Less than 12 months | 27 (6.59%) | 306 (74.635) | 333 (81.22%) |
| | 12–24 month | 12 (2.93%) | 25 (6.10%) | 37 (9.02%) |
| | Above 24 months | 7 (1.71%) | 33 (8.05%) | 40 (9.76%) |
| **Active TB** | No | 25 (6.10) | 278 (67.80%) | 303 (73.90%) |
| | Yes | 21 (5.12%) | 86 (20.98%) | 107 (26.10%) |
| **Active Hepatitis B virus** | Negative | 39 (9.51%) | 349 (85.12%) | 388 (94.63%) |
| | Positive | 7 (1.71%) | 15 (3.66%) | 22 (5.37%) |
| **HIV support group care utilization** | No | 41 (10.00%) | 264 (64.39%) | 305 (74.39%) |
| | Yes | 5(1.22%) | 100 (24.39%) | 105 (25.61%) |

## Determinants of virological outcomes of antiretroviral therapy

In multivariate logistic regression analysis, wealth level, active TB, hepatitis B virus, and HIV support group care utilization were found to be significant determinants of virological outcomes of antiretroviral therapy. The odds of viral suppression were 2.67 times higher among individuals in the rich household wealth category compared to those in the poor category (AOR = 2.67, 95% CI: 1.15–6.22). Individuals with active TB and hepatitis B virus were 73% (AOR = 0.27, 95% CI: 0.14–0.57) and 80% (AOR = 0.20, 95% CI: 0.10–0.31) less likely to achieve viral suppression, respectively, compared to those without these conditions. HIV patients who received HIV support group care were 3.14 times more likely to achieve viral suppression than those who did not (AOR = 3.14, 95% CI: 1.35–6.30) *(Table 4)*.

## Discussion

This study aimed to investigate the virological outcomes and their determinants among HIV patients. The findings revealed that 364 participants (88.78%) successfully achieved viral

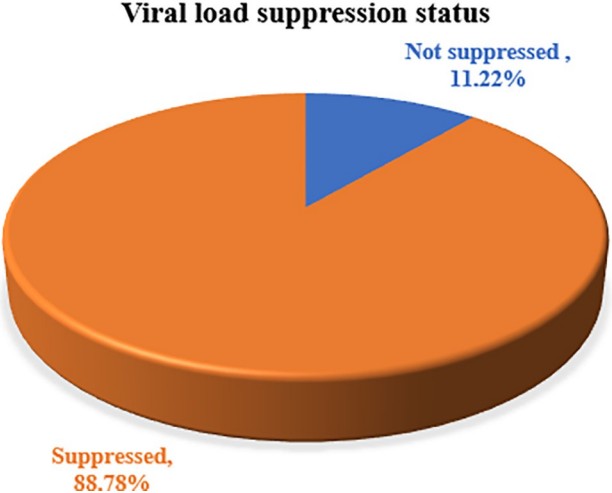

**Fig 1. Viral suppression status of antiretroviral therapy among HIV patients in Ethiopia.**

suppression. Significant determinants of virological outcomes included wealth level, active TB, hepatitis B virus, and utilization of HIV support group care.

In this study, the prevalence of viral suppression was 88.78%, which falls short of the WHO's target of 95% by 2025 [19]. The finding highlights the need for better adherence support, optimized ART regimens, and strengthened healthcare systems. It also emphasizes the importance of addressing barriers to care and developing targeted public health policies to meet the WHO's suppression goals. The findings is slightly lower than the study done in Nepal (90.08%) [20], in African cohort (91%) [21] and Vietnam (93%) [22]. In contrast, this finding is higher than the 76.1% viral suppression rate reported in Ghana, 82% in Botswana, 80.7% in Uganda, and 59% in Kenya [23–26]. The variation in viral suppression rates across different studies and regions can be attributed to several key factors. Countries with advanced healthcare systems, such as Vietnam, typically achieve higher suppression rates due to better access to effective antiretroviral therapy (ART) and comprehensive support services. In contrast, regions with less developed healthcare infrastructure may face challenges that affect ART

**Table 3. Multicollinearity test to examine the relationship between explanatory variables.**

| Variable | VIF | 1/VIF |
|---|---|---|
| **Sex** | 1.17 | 0.857918 |
| **Age in year** | 1.11 | 0.901931 |
| **Wealth level** | 1.11 | 0.903844 |
| **Education level** | 1.09 | 0.917067 |
| **Marital status** | 1.09 | 0.918469 |
| **ART duration** | 1.05 | 0.953809 |
| **HIV support group care utilization** | 1.05 | 0.955170 |
| **Active TB** | 1.04 | 0.958362 |
| **Alcohol consumption** | 1.03 | 0.974990 |
| **Time to reach to HIV care service** | 1.03 | 0.974993 |
| **Active Hepatitis B virus** | 1.02 | 0.976898 |
| **Mean VIF** | 1.07 | |

**Table 4. Binary and multivariate logisitic regression analysis of determinants of virological outcomes of antiretroviral therapy among HIV patients in Ethiopia.**

| Variables | Categories | CHR (95CI) | AHR (95CI) | P-value |
|---|---|---|---|---|
| **Age in year** | <25 | 1 | 1 | |
| | 25–45 | 2.38(0.95–5.98) | 3.83(0.89–6.74) | 0.068 |
| | >45 | 6.36(1.71–13.73) | 8.45(0.93–11.54 | 0.087 |
| **Sex** | Male | 1 | 1 | |
| | Female | 0.46(0.19–1.13) | 3.08(0.91–5.67) | 0.145 |
| **Marital status** | Married | 1 | 1 | |
| | Not married | 0.55(0.29–1.05) | 0.87(0.43–1.79) | 0.721 |
| **Education level** | No education | 1 | 1 | |
| | Primary | 1.39(0.65–3.00) | 1.11(0.46–2.68) | 0.810 |
| | Secondary and above | 1.18(0.49–2.83) | 0.95(0.3–2.67) | 0.924 |
| **Wealth level** | Poor | 1 | 1 | |
| | Middle | 2.51(1.11–5.67) | 1.94(0.78–4.81) | 0.152 |
| | Rich | 2.39(1.17–4.85) | **2.67(1.15–6.22)** | **0.002** |
| **Alcohol consumption** | Yes | 1 | **1** | |
| | No | 0.98(0.51–1.91) | 1.19(0.57–2.47) | 0.647 |
| **Time to reach to HIV care service** | Less than 1 hr. | 1 | 1 | |
| | 1hr. and above | 1.34(0.55–3.30) | 1.11(0.42–2.96) | 0.825 |
| **ART duration** | Less than 12 months | 1 | 1 | |
| | 12–24 month | 1.69(0.32–6.11) | 1.48(0.23–4.40) | 0.676 |
| | Above 24 months | 1.81(0.75–4.37) | 1.02(0.37–2.81) | 0.974 |
| **Active TB** | No | 1 | 1 | |
| | Yes | 0.37(0.19–0.69) | **0.27(0.14–0.57)** | **0.000** |
| **Active Hepatitis B virus** | Negative | 1 | 1 | |
| | Positive | 0.24(0.11–0.43) | **0.20(0.10–0.31)** | **0.005** |
| **HIV support group care utilization** | No | 1 | **1** | |
| | Yes | 3.10(1.19–5.42) | **3.14(1.35–6.30)** | **0.008** |

effectiveness and adherence. Additionally, socioeconomic differences, variations in ART protocols, and differences in study populations also contribute to these disparities.

Household wealth level was found to be a significant determinant of virological suppression, with HIV-positive individuals from poor wealth index levels being less likely to achieve viral suppression compared to those in wealthier categories. This findings in agreement previous studies [27–29]. The finding that household wealth level significantly affects virological suppression highlights the need for targeted interventions and equitable resource allocation to support individuals from poorer backgrounds. It underscores the importance of addressing socioeconomic barriers to enhance ART adherence and improve health outcomes. Public health policies and programs should be designed to meet the specific needs of lower-income populations to achieve better virological outcomes and reduce disparities in HIV care.

Having active TB and hepatitis B virus significantly reduces the viral suppression rate. Similarly, other studies have reported comparable findings [30–33], indicating that co-infections can adversely impact the effectiveness of antiretroviral therapy and viral suppression outcomes. Active TB and hepatitis B virus significantly reduce viral suppression by complicating HIV treatment and potentially interfering with the effectiveness of antiretroviral therapy (ART). These co-infections can exacerbate the progression of HIV, hinder adherence to ART, and reduce the overall efficacy of the treatment, highlighting the need for integrated management strategies that address both HIV and co-infections to improve treatment outcomes.

Effective management of co-infections is crucial for improving ART outcomes and achieving better viral suppression. Public health strategies should include screening and treatment for co-infections alongside HIV care.

This finding highlights that HIV support group care is crucial for achieving viral suppression. Individuals who participate in HIV support groups are three times more likely to achieve viral suppression compared to those who do not. This association between HIV support care and improved viral suppression is also consistent with findings from previous studies [34, 35]. Support group services encompass a range of supportive measures, including counseling and health living messages, reminders about the importance of taking antiretroviral medications (ARVs) regularly, and prompts to keep HIV-related appointments. They also assist with refilling or picking up ART medication, provide psychosocial support, and offer livelihood and material support [36]. Integrating support group services into HIV care programs can provide essential emotional and practical support, which can lead to better adherence to antiretroviral therapy (ART) and improved health outcomes. This suggests that expanding access to support groups and incorporating them as a standard part of HIV care could be an effective strategy for increasing viral suppression rates and overall treatment success.

Since the Population-based HIV Impact Assessment Survey was conducted exclusively in urban areas, this study does not address viral suppression rates in rural settings. The focus on urban populations limits the generalizability of the findings to rural contexts, where different challenges and factors might influence HIV care and treatment outcomes. Additionally, the cross-sectional design of the data restricts the ability to infer causal relationships between variables. Furthermore, as the study relies on secondary data analysis, it is constrained by the limitations inherent in the original data collection process. This means that while the study can explore associations, it cannot investigate other potential determinants of virological suppression that may not have been included or detailed in the original survey. Consequently, further research is needed to address these gaps and explore additional factors influencing viral suppression, particularly in different settings and with more detailed longitudinal data.

## Conclusions

In conclusion, this study reveals that the prevalence of viral suppression among HIV patients is 88.78%, which even falls short of the WHO's 90% target for 2020. This shortfall underscores several critical determinants influencing virological outcomes. Key factors identified include household wealth level, the presence of active TB and hepatitis B virus, and the utilization of HIV support group care. The study highlights the urgent need for targeted interventions to address these factors. Improving support for low-income individuals, who are less likely to achieve viral suppression, is essential. Additionally, effective management of co-infections such as TB and hepatitis B virus is crucial for enhancing ART outcomes. Expanding access to and utilization of HIV support group services can provide significant benefits, including adherence support and psychosocial assistance. Addressing these determinants through comprehensive strategies will be vital in closing the gap between current suppression rates and the WHO's target. Such efforts are critical for improving overall health outcomes and achieving better viral suppression rates for people living with HIV.

## Acknowledgments

The author is indebted to the PHIA Project for giving permission to access the dataset.

## Author Contributions

**Conceptualization:** Tegene Atamenta Kitaw.

**Data curation:** Tegene Atamenta Kitaw.

**Formal analysis:** Tegene Atamenta Kitaw, Ribka Nigatu Haile.

**Methodology:** Tegene Atamenta Kitaw, Ribka Nigatu Haile.

**Resources:** Tegene Atamenta Kitaw.

**Software:** Tegene Atamenta Kitaw.

**Supervision:** Tegene Atamenta Kitaw, Ribka Nigatu Haile.

**Writing – original draft:** Tegene Atamenta Kitaw.

**Writing – review & editing:** Tegene Atamenta Kitaw, Ribka Nigatu Haile.

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
