## [Decision Letter · Decision Letter 0]

25 Oct 2024

Virological Outcomes of Antiretroviral Therapy and its Determinants Among HIV Patients in Ethiopia: Implications for Achieving the 95–95–95 Target.

PONE-D-24-29997

Dear Dr. Kitaw,

We’re pleased to inform you that your manuscript has been judged scientifically suitable for publication and will be formally accepted for publication once it meets all outstanding technical requirements.

Kind regards,

Justyna Dominika Kowalska

Academic Editor

PLOS ONE

Journal Requirements:

1. Your ethics statement should only appear in the Methods section of your manuscript. If your ethics statement is written in any section besides the Methods, please move it to the Methods section and delete it from any other section. Please ensure that your ethics statement is included in your manuscript, as the ethics statement entered into the online submission form will not be published alongside your manuscript.

Reviewers' comments:

Reviewer's Responses to Questions

**Comments to the Author**

1. Is the manuscript technically sound, and do the data support the conclusions?

Reviewer #1: Yes

Reviewer #2: Yes

2. Has the statistical analysis been performed appropriately and rigorously? 

Reviewer #1: Yes

Reviewer #2: Yes

3. Have the authors made all data underlying the findings in their manuscript fully available?

Reviewer #1: Yes

Reviewer #2: Yes

4. Is the manuscript presented in an intelligible fashion and written in standard English?

Reviewer #1: Yes

Reviewer #2: Yes

5. Review Comments to the Author

Reviewer #1: The result that there is relation between household wealth level, and viral suppression rates is very interesting, but in manuscript should be indicate what is the major funding resources of ARV for studied patients with HIV. In some countries this treatment is finansed by goverment, and propably this result can not be generalized for all patients with HIV. When you have access to free treatment, there can be other factors related to household than only financial aspects, so it will be interesting to find them in next studies. But in this study there is lack of information on what is relation of this housholds with needs of ARV treatment finansing by patients.

Also focus only on urban population can be treated as limitation of this study, but authors are aware of it.

So in the next steps in the future it is recommended to expand this study to population from different place of residence.

Reviewer #2: The research topic is timely and relevant, appropriate methodology and proper statistical analysis with detailed discussion.

Overall it is well written and informative article. Including participants from urban areas only might limit the research generalization, since most of our populations are in rural areas.

6. PLOS authors have the option to publish the peer review history of their article (what does this mean?). If published, this will include your full peer review and any attached files.

Reviewer #1: No

Reviewer #2: No

---

## [Editor Report · Acceptance letter]

17 Dec 2024

PONE-D-24-29997 

PLOS ONE

Dear Dr. Kitaw, 

I'm pleased to inform you that your manuscript has been deemed suitable for publication in PLOS ONE. Congratulations! Your manuscript is now being handed over to our production team.

Kind regards, 

on behalf of

Prof. Justyna Dominika Kowalska 

Academic Editor

PLOS ONE